# A Validation Study of the Korean Version of the Nurses’ Patient Education Questionnaire

**DOI:** 10.3390/ijerph18115609

**Published:** 2021-05-24

**Authors:** Myung-Jin Jung, Young-Sook Roh

**Affiliations:** 1College of Nursing, Korea University, Seoul 02841, Korea; malon175@cau.ac.kr; 2Red Cross College of Nursing, Chung-Ang University, Seoul 06974, Korea

**Keywords:** factor analysis, hemodialysis, nurse, patient education, reliability

## Abstract

This study aimed to investigate the internal consistency reliability and construct validity of the Korean version of the Nurses’ Patient Education Questionnaire. An accurate assessment of nurses’ perceived patient education competency is required, and these needs assessment results can provide the evidence for designing a continuing education to empower and equip nurses with optimal competency for patient education. A cross-sectional study was used to which a convenience sample of 262 hemodialysis unit nurses in the Republic of Korea. In the exploratory factor analysis, the questionnaire consisted of 26 items belonging to seven factors: (1) nurse belief and knowledge; (2) documentation of patient education activities; (3) collegial teamwork; (4) educational environment; (5) interdisciplinary cooperation; (6) education foundation; and (7) health care organization, all of which accounted for 70.2% of the variance. The internal consistency reliability was 0.91 for the overall scale and each factor at 0.70 to 0.77. The questionnaire provides a valid and reliable instrument to assess nurses’ patient education competency.

## 1. Introduction

The prevalence of limited health literacy among hemodialysis patients ranged from 10% to 50% [1], and limited health literacy was associated with nonadherence, adverse events, and mortality [2]. Patient education can help patients implement patient adherence activities and induce positive physiological and psychosocial patient outcomes [3,4]. However, patient education for patients with end-stage renal disease undergo hemodialysis can be challenging and competing with all other topics for hemodialysis unit nurses because they face patient, nurse, and hospital-related barriers for implementing optimal patient education [5,6]. Patient education competency is one of the most important factors for nurses’ educational performance [7]. Nurses’ patient education competency is also known as one of the nurses’ professional competency items [8,9,10,11]. As educators, nurses play an important role in patient education and they require varied communicational, educational, animation, and assessment skills [12,13]. 

Patient education is aimed to develop patient competence, confidence, and self-trust in their ability to carry out health behaviors consistent with their life plan [14]. The concept of patient education has shifted from patient teaching to patient empowerment [15] and shared decision making [16]. Therapeutic patient education is important for nurses who work with patients with chronic diseases because it helps patients and their families acquire or maintain the skills to improve adherence and quality of life with a chronic disease by providing psychosocial support and information [13,17]. Therapeutic patient education competency includes knowledge, know-how, and educative attitude [18]. The main factors required to become an expert educator included a supportive learning environment, inner motivation, and an awareness of the value of patient education [19]. Healthcare professionals reported that knowledge, communication skills, and pedagogical competencies are required for evidence-based and high-quality patient education in patients with coronary heart disease [20]. The patients perceived the capability of building trust and tailoring the education to the individual needs and situations as the most prominent characteristics of a good educator [21].

An accurate assessment of nurses’ perceived patient education competency is required, and these needs assessment results can provide the evidence for designing a continuing education to empower and equip nurses with optimal competency for patient education. Several tools have been used to measure the patient education competency of nurses [22,23,24]. However, the content of each item is rather abstract and focuses on the evaluation of psychometric characteristics of nursing students and entry-level nurses [23], which is not enough to explain the comprehensive perception of patient education by clinical nurses. The 24-item Patient Education Competence Scale for Registered Nurses (PECS-RN) is limited to two factors and only emphasizes the relevance of clinical teaching capabilities [24]. 

The Nurses’ Patient Education Questionnaire (NPEQ) measures ‘what the nurses do in practice’ and attitude ‘what they think about what they do’ as a prerequisite for a nurse’s patient education activities [25]. The conceptual framework of the NPEQ was developed through integrative review [26]. It is a comprehensive tool for studying conditional factors based on the recognition of important matters affecting nurses’ performance of patient education and the factors that provide important information. It was tested on 701 nurses (83% response rate) in Sweden to evaluate nurses’ awareness of patient education in five areas of 47 items [25]. However, the construct validity of the tool has not been verified. The purpose of this study was to verify the reliability and validity of the NPEQ [25] in Korean hemodialysis unit nurses. The Korean version of the Nurses’ Patient Education Questionnaire (NPEQ-K) can be used to assess the current status and training needs of nurses to improve nurses’ patient education competency.

## 2. Materials and Methods

### 2.1. Design

The present study employed a descriptive and cross-sectional survey design.

### 2.2. Participants

The minimum required sample size was estimated based on the suggestion that there be at least 200 participants for exploratory factor analysis [27]. The criteria for selecting the study participants were registered nurses working in the hemodialysis units of private clinics, general hospitals, and advanced general hospitals in six provinces of the Republic of Korea. The hemodialysis units were selected according to regional allocation and classifications of hospital types using a convenience sampling method. A convenience sample of 268 Korean hemodialysis unit nurses was recruited from 27 hemodialysis units in six provinces of the Republic of Korea after considering the 30% dropout rate. The final sample size was 262, which satisfied the minimum number of participants for exploratory factor analysis.

### 2.3. Measurements

#### 2.3.1. General Characteristics

The general characteristics of nurses were measured using a questionnaire. The questionnaire included 12 questions concerning demographic characteristics (age, gender, academic degree, hospital type, position, nurse total career, working experience in hemodialysis unit, number of nurses in hemodialysis unit, position, working style, number of patients per nurse, number of assistant personnel) and 9 questions regarding patient education-related characteristics (access to patient education materials, presence of education nurse, presence of patient education booklet, presence of regular education programs, frequency of patient education within the last one week, pedagogical experience, types of training materials, educated institutions, place for patient education) [28].

#### 2.3.2. Nurses’ Patient Education Competency

Nurses’ patient education competency was measured using the Nurses’ Patient Education Questionnaire. The initial items were developed based on the literature review and validated by a group of experts [25,28]. First, the authors obtained permission from the tool developer to use the tool, including the Korean translation. In order to ensure the quality and fidelity of the Korean version of NPEQ, the authors received descriptions and advice from the first developers of NPEQ via email. The English version of the tool was translated by the authors and then translated into the Korean version using a translation-reverse translation method through a translation agency.

The content validity index was evaluated by six content experts (two professors of nursing and four hemodialysis unit nurses with more than 20 years of clinical experience using the Content Validity Index. The experts rated the adequacy and clarity of each item by using a four-point Likert scale (1 = “not appropriate,” 2 = “somewhat appropriate,” 3 = “appropriate,” and 4 = “very appropriate”). Some wording revisions were made according to the experts’ suggestions. The individual item CVI was computed as the proportion of content experts giving an item a relevancy rating of 3 or 4. The results of the item-level CVIs ranged from 0.83 to 1; the CVI of the overall scale was 0.97. The authors finalized the 33 items with more than 80% agreement in item level content validity index according to Lynn’s criteria [29]. Three hemodialysis unit nurses participated in a pilot test of the Korean version Nurses’ Patient Education Questionnaire. No items were revised during this process.

Nurses reported their competency using a 5-point Likert scale ranging from 1 = never to 5 = always or 1 = strongly disagree to 5 = strongly agree. The higher the total score, the higher the perceived competency. The Cronbach’s alpha coefficient for the original tool was 0.97 [28], and 0.91 for this study.

### 2.4. Data Collection

Referring to the classification of regional allocation and hospital type, the nursing departments of hospitals located in six provinces of the Republic of Korea were contacted in advance to explain the purpose of the study and to request data collection. The questionnaire was then distributed by the nurse of the nursing department to hemodialysis unit nurses who agreed to participate. Data were collected from 1 October 2018 to 5 December 2018. The completed questionnaire was collected in a batch by the education nurse of the nursing department and either directly visited by the researcher or received by mail. As a compensation for participating in the study, a coffee coupon worth 5USD was provided to participants.

### 2.5. Data Analysis

The data were analyzed using IBM SPSS Statistics for Windows, Version 23.0 (IBM Corp., Armonk, NY, USA). Cronbach’s alpha coefficients were calculated to assess internal consistency reliability. Exploratory factor analysis was carried out using principal component analysis with varimax rotation. Known group analysis was performed using two sample *t*-tests.

## 3. Results

### 3.1. General Characteristics

A total of 268 questionnaires were distributed and 264 were returned (98.5%). Data from 262 were used for the final analysis, with two participants excluded due to incomplete data. Table 1 shows the demographic characteristics of nurses. Regarding the characteristics related to patient education of hemodialysis nurses, 90.5% (*n* = 237) have access to educational materials, 49.6% (*n* = 130) have access to a dedicated patient education nurse, 88.2% (*n* = 231) have patient education booklets, and 58.0% (*n* = 152) have regular patient education programs. Of the participants, 71.8% (*n* = 188) of nurses had pedagogical experience (yes), and frequency of patient education experience within the last 1 week was 4.99 ± 11.58. The most commonly used type of patient education materials was hospital booklet 26.7%, followed by oral education (24.0%), and pharmaceutical brochure (23.3%). The most commonly cited institution that received education in pedagogy was college (36.7%). The main place to educate patients (multiple response) was in the patient’s room with the presence of other patients 47.1% (Table 1).

### 3.2. Descriptive Statistics of the Items

The item means and standard deviations, inter-item correlation matrix, and item–total correlations were computed and examined. The item–to–total scale correlations of the preliminary 33-item scale ranged from 0.29 to 0.71. Three items were eliminated because of redundancy or lack of homogeneity with the construct: (1) I am uncertain of what patient information the physician should provide as opposed to myself as a nurse and what the content should be (0.29); (2) I am uncertain of what patient teaching the physician should provide as opposed to myself as a nurse and what the content should be (0.28); and (3) I know how to document patient teaching in the record (0.71). The remaining 30 items had corrected item–total correlation coefficients between 0.31 and 0.68 which is an acceptable range [30]. 

### 3.3. Construct Validity

Prior to performing the factor analysis, the suitability of the data for factor analysis was assessed. Bartlett’s test of sphericity was significant (χ^2^ = 3875.002, *p* < 0.001), and the KMO measure of sampling adequacy was appropriate at 0.857. An exploratory factor analysis identified seven factors based on the 26 items with >0.6 factor loading, which is considered suitable for factor analysis [27]: nurses’ beliefs and knowledge (6 items), documentation of patient education activities (5 items), collegial teamwork (4 items), education environment (4 items), interdisciplinary cooperation (3 items), educational foundation (2 items), and health care organization (2 items). All items had acceptable factor loadings, ranging from 0.60 to 0.93. Four items with <0.60 factor loading were deleted: (1) I know how to document patient information in the record (0.40); (2) I document learning objectives for patient teaching in the record (0.47); (3) After the patient has had a conversation with the physician, I check how the patient has understood the physician’s patient education (0.45); (4) I think patient teaching is an important nursing responsibility (0.45). Together, these 26 items accounted for 70.2% of the variance (Table 2).

### 3.4. Reliability

On a 5-point scale, the overall mean score was 3.35 ± 0.47. The highest average was educational foundation (3.90 ± 0.65), and the lowest factor was health care organization (3.13 ± 0.77). The reliability estimates of the seven factors ranged from 0.70 to 0.77. The instrument demonstrated high internal consistency, with an alpha value of 0.91. The correlations between the seven factors ranged from 0.300 to 0.647 (Table 2).

### 3.5. Known Group Analysis

New graduate nurses reported lower nurse patient education competency scores overall (t = −3.21, *p* = 0.001) and its subscales compared to experienced nurses: nurses’ beliefs and knowledge (t = −3.59, *p* < 0.001), collegial teamwork (t = −3.95, *p* < 0.001), interdisciplinary cooperation (t = −2.45, *p* = 0.015). However, there were no significant differences in documentation of patient education activities (t = −1.78, *p* = 0.077), education environment (t = −0.836, *p* = 0.404), educational foundation (t = 0.259, *p* = 0.796), and health care organization (t = −0.674, *p* = 0.518) (Table 3).

## 4. Discussion

Our findings demonstrate the acceptable internal consistency reliability and construct validity of the Nurses’ Patient Education Questionnaire (NPEQ) in Korean hemodialysis unit nurses. Exploratory factor analysis identified seven factors, termed “nurses’ beliefs and knowledge, documentation of patient education activities, collegial teamwork, education environment, interdisciplinary cooperation, educational foundation, and health care organization.” These seven factors accounted for 70.2% of the variance, meeting the threshold of 60% or higher variance explained [31]. The 18-item Health Education Competency Scale explained 75.9% variance [23], and the 24-item Patient Education Competence Scale accounted for 85.0% of the variance [24]. The factors identified in our study are similar to those identified in previous studies on patient education competency [18,20,26]. Furthermore, nurses reported the highest perception in educational foundation, and the health care organization was the lowest. Therefore, nurse educators need to design and implement continuing education for competency enhancement by identifying the actual competency level of nurses according to the attributes of nurse patient education competency in the present study. There is a need to develop strategies for enhancing health care organizational support to implement patient education by nurses.

The seven-factor NPEQ showed satisfactory internal consistency, with a total Cronbach’s alpha of 0.91 above the recognized threshold of 0.70 [32]. Cronbach’s alpha ranged from 0.90 to 0.96 for the Health Education Competency Scale [23]. The Patient Education Competence Scale for Registered Nurses showed the Cronbach’s α of 0.97 for the “implementation of patient education” subscale, 0.96 for the “preparation for patient education” subscale, and 0.98 for the overall scale [24]. Therefore, the NPEQ is a reliable tool to measure nurse patient education competency.

The initial NPEQ was a 33-item tool that included questions on the five domains assessing nurses’ beliefs and knowledge; (1) education environment; (2) health care organization; (3) interdisciplinary cooperation; (4) collegial teamwork; and (5) patient education activities [25] which we systematically refined to 26 items in the present study. To date, few measures have been used in nurse patient education studies, which may raise the issue of validity. The 26-item NPEQ-K in our study reflects the perspective and characteristics of hemodialysis unit nurses as a patient educator.

“Nurses’ beliefs and knowledge” was found to be the most important factor for the nurse patient education competency. Nurses’ beliefs and knowledge include nurse’s attitude, responsibility, and personal competence/teaching skills toward patient education [25]. Pétré et al. (2017) also reported that therapeutic patient education competency includes knowledge, know-how, and educative attitude. Patients, system, and provider-related factors are suggested as barriers to optimal patient education [33]. Among them, the lack of knowledge and awareness of nurses was suggested as a major nurse-related factor [5]. In order to induce positive outcomes of patient education interventions, it is necessary to strengthen competency by grasping the knowledge and beliefs of nurses about patient education and identifying areas for improvement. Nurses’ motivation for patient education can be improved by supervising patient education and providing constructive feedback and empowering the nursing profession [12]. Nurse educators need to design and implement strategies to improve nurses’ knowledge, beliefs, and motivation for patient education.

The present study found that new graduate nurses reported lower nurse patient education competency scores in overall and its subscales compared to experienced nurses. In particular, new graduate nurses showed lower perceived competency than experienced nurses in the nurses’ beliefs and knowledge, collegial teamwork, and interdisciplinary cooperation. Given the reality that patient education is not a priority due to competing work demands and the missing workplace culture to teach [34], new graduate nurses may face various difficulties in patient education. As new graduate nurses lack practical experience in patient education, nurse educators need to provide continuing education programs to strengthen their competency [12]. In addition, it is necessary to develop and implement related subjects in the undergraduate curriculum so that nursing students can acquire knowledge, skills, and attitudes about patient education.

The strength and major contribution of this study is its establishment of satisfactory internal consistency reliability and construct validity in the NPEQ-K. However, there are some limitations need to be considered. As this research was conducted among Korean hemodialysis unit nurses, it offers a limited scope for generalization in other countries. A further refinement of the instrument could benefit the nurse patient education research area. As this tool is a self-administered tool to identify the nurses’ perception on patient education competency, the subjectivity of respondents cannot be excluded.

## 5. Conclusions

Nurse patient education competency can be defined as nurses’ knowledge, skills, and attitude to design, implement, and evaluate patient education to promote health literacy, patient adherence, and the empowerment of patients. This study provided evidence to support the effectiveness and reliability of NPEQ for patient education of hemodialysis nurses in Korea. Because the adequacy and application of the instrumentation have been considered in the configuration of NPEQ-K, future strategies to identify the types of needs that are important to nurses’ patient education in practice and further to manage and improve the impact on nurses’ patient education performance can be designed using NPEQ-K. Further study is necessary to develop and implement related curriculum so that nursing students or new graduate nurses can acquire knowledge, skills, and attitudes about patient education. It is needed to identify the effects of a continuing education program on nurses’ patient education competencies.

## Figures and Tables

**Table 1 ijerph-18-05609-t001:** General characteristics of nurses (*n* = 262).

Variable	Category	*n* (%)	Mean ± SD
Age (years)	22–29	62 (23.7)	36.91 ± 8.82
	30–39	98 (37.4)	
	40–49	63 (27.8)	
	≥50	29 (11.1)	
Gender	Man	1 (0.4)	-
	Woman	261 (99.6)	
Academic degree	Associate	70 (26.7)	-
	Bachelor	161 (61.5)	
	Master	30 (11.5)	
	Doctor	1 (0.4)	
Hospital type	Private clinic	81 (30.9)	-
	General hospital	134 (51.1)	
	Advanced general Hospital	47 (17.9)	
Position	Head nurse	20 (7.6)	-
	Charge nurse	41 (15.6)	
	Staff nurse	201 (76.7)	
Nurse total career (year)			12.72 ± 8.52
Working experience in hemodialysis unit (year)			6.45 ± 5.94
Number of nurses in hemodialysis unit (including head nurse)			13.44 ± 6.04
Number of patients per nurse			6.87 ± 1.89
Number of assistant personnel			1.47 ± 1.31
Access to educational materials (yes)		237 (90.5)	
Presence of education nurse (yes)		130 (49.6)	
Presence of patient education booklet (yes)		231 (88.2)	
Presence of regular education programs (yes)		152 (58.0)	
Pedagogical experience (yes)		188 (71.8)	
Frequency of patient education experience within the last 1 week			4.99 ± 11.58
Types of training materials (multiple response)	Oral education	141 (24.0)	
	Hospital booklet	157 (26.7)	
	Pharmaceutical brochure	137 (23.3)	
	Clinical management guidelines	113 (19.2)	
	Patient education application program	19 (3.2)	
	Web program site	13 (2.2)	
	Video and imaging	6 (1.0)	
	Power point template	2 (0.3)	
Educated institutions (multiple response)	College	116 (36.7)	
	Conference	68 (21.5)	
	Continuing education	62 (19.6)	
	Hospital education	45 (14.2)	
	Graduate study	23 (7.3)	
	Self-study	2 (0.6)	
Place for patient education (multiple response)	Patient’s room with the presence of other patients	254 (47.1)	
	Nursing office	84 (15.6)	
	Corridor	74 (13.7)	
	Private room/patient’s home	33 (6.1)	
	Patients’ dining room at the ward	16 (2.9)	
	Medical office	5 (0.1)	
	Others	78 (14.5)	

**Table 2 ijerph-18-05609-t002:** Factor analysis of 26-item of the instrument (*n* = 262).

Item	Factor						
**Factor I. Nurses’ beliefs and knowledge**	**I**	**II**	**III**	**IV**	**V**	**VI**	**VII**
1. I am qualified/competent in patient information work.	**0.795**	0.217	0.118	0.141	0.178	0.033	−0.154
2. I am qualified/competent in patient teaching work.	**0.793**	0.160	0.123	0.108	0.280	0.022	−0.133
3. I follow the development of patient education knowledge in scientific literature, e.g., articles in journals.	**0.776**	0.210	0.137	0.121	−0.032	0.103	0.211
4. I follow the development of patient education knowledge in professional literature, e.g., books and professional journals.	**0.765**	0.158	0.108	0.038	−0.066	0.157	0.254
5. I know how to document the patient’s need for knowledge about the health situation in the record.	**0.665**	0.321	0.181	0.052	0.033	0.046	0.151
6. I know what my mandate is in patient teaching and information.	**0.615**	0.175	0.232	0.056	0.101	0.100	0.101
**Factor II.** **Documentation of patient education activities**							
7. I document nursing evaluation of patient information in the record.	0.171	**0.764**	0.212	0.107	0.042	0.048	0.210
8. I document nursing activities of patient teaching in the record.	0.323	**0.760**	0.197	0.107	0.063	−0.059	0.009
9. I document nursing evaluation of patient teaching in the record.	0.259	**0.736**	0.234	0.133	0.025	−0.029	0.287
10. I document nursing activities of patient teaching in the record.	0.162	**0.722**	0.187	0.086	0.165	0.058	−0.164
11. I document the patients’ need of knowledge of their health situation in the record.	0.270	**0.679**	0.072	0.262	−0.054	0.064	0.131
**Factor III. Collegial teamwork**							
12. At my workplace, we discuss nurses’ patient information, e.g., activity/area information.	0.083	0.193	**0.821**	0.072	0.062	0.019	0.148
13. At my workplace, we discuss nurses’ patient teaching, e.g., activity/area teaching.	0.260	0.147	**0.798**	0.114	−0.005	−0.035	0.155
14. At my workplace, I discuss with colleagues how to help the patients acquire knowledge.	0.147	0.249	**0.779**	0.062	0.105	0.028	0.098
15. I make sure I know about the patients’ need of knowledge in terms of what the patients want to know and understand about their health situation (directly from the patient).	0.329	0.176	**0.614**	0.143	0.156	0.081	0.040
**Factor IV. Education environment**							
16. I can teach the patient undisturbed, e.g., I am not disturbed by colleagues, other patients or by phone calls.	0.055	0.159	−0.075	**0.857**	0.013	0.093	0.174
17. I can inform the patient undisturbed, e.g., I am not disturbed by colleagues, other patients or by phone calls.	0.031	0.135	0.007	**0.799**	0.002	0.000	0.276
18. I have time for patient teaching in my daily work.	0.155	0.091	0.223	**0.718**	0.066	0.059	−0.104
19. I have time for patient information in my daily work.	0.168	0.163	0.324	**0.691**	0.148	0.019	−0.142
Factor V. Interdisciplinary cooperation							
20. Co−operating with other professionals in patient education is important.	0.052	0.111	0.120	0.012	**0.929**	0.101	−0.042
21. It is important that the nurse is responsible for co−organizing patient education between different professional groups in regard to the patients the nurse is caring for/nursing.	0.107	0.042	0.141	−0.012	**0.907**	0.315	0.038
22. Patient teaching has high priority in my daily work.	0.157	0.000	−0.025	0.203	**0.638**	0.072	0.302
**Factor VI. Educational foundation**							
23. To meet patient teaching needs, it is important that I am knowledgeable in the following subjects: medical science, nursing/caring science, educational science and psychological science.	0.149	0.019	0.044	0.067	0.172	**0.934**	0.008
24. To meet patient information needs, it is important that I am knowledgeable in the following subjects: medical science, nursing/caring science, educational science and psychological science.	0.127	0.030	0.001	0.082	0.239	**0.909**	0.022
**Factor VII. Health care organization**							
25. I have support from my manager in my patient teaching/information, i.e., the activity patient teaching.	0.023	0.208	0.238	0.126	0.120	0.072	**0.721**
26. My manager offers professional development in the activity/area of patient education.	0.395	0.070	0.207	0.057	0.098	−0.063	**0.602**
Mean ± SD	3.20 ± 0.68	3.21 ± 0.78	3.38 ± 0.69	3.18 ± 0.69	3.83 ± 0.54	3.90 ± 0.65	3.13 ± 0.77
Cronbach’s α	0.702	0.706	0.713	0.744	0.747	0.773	0.733
Eigenvalue	8.311	2.412	2.119	1.751	1.372	1.204	1.083
Percent of total variance explained	15.622	12.520	11.213	10.195	7.447	7.334	5.869
Cumulative percent	15.622	28.142	39.355	49.550	56.997	64.331	70.200

Bold entries indicate items included in the respective factor.

**Table 3 ijerph-18-05609-t003:** Comparison of Scores between New Graduate and Experienced Nurses.

Factor	New Graduate Nurses * (*n* = 42)	Experienced Nurses (*n* = 220)	t	*p*
	Mean ± SD			
I. Nurses’ beliefs and knowledge	2.86 ± 0.66	3.26 ± 0.67	−3.59	<0.001
II. Documentation of patient education activities	3.01 ± 0.75	3.25 ± 0.78	−1.78	0.077
III. Collegial teamwork	3.01 ± 0.70	3.45 ± 0.66	−3.95	<0.001
IV. Education environment	3.10 ± 0.63	3.19 ± 0.70	−0.836	0.404
V. Interdisciplinary cooperation	3.64 ± 0.54	3.87 ± 0.54	−2.45	0.015
VI. Educational foundation	3.93 ± 0.61	3.90 ± 0.66	0.259	0.796
VII. Health care organization	3.06 ± 0.73	3.14 ± 0.77	−0.674	0.518
Total	3.14 ± 0.41	3.39 ± 0.47	−3.21	0.001

* divided by work duration, new graduate nurses: less than 12 months, experienced nurses: more than 13 months.

## Data Availability

The data presented in this study are available on request from the corresponding author. The data are not publicly available due to the information contained that could compromise the privacy of research participants.

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
