# Peer review of "A Validation Study of the Korean Version of the Nurses’ Patient Education Questionnaire"

_ijerph, 2021, doi:10.3390/ijerph18115609_

Round 1

Reviewer 1 Report

Congratulate the authors for the great work presented to the magazine. After a critical reading in depth. I only have one question, regarding the calculation of the sample and the choice of the service, since methodologically the work is impeccable.

Author Response

Reviewer# 1: Comments to Author:

Author response to comments:

Congratulate the authors for the great work presented to the magazine. After a critical reading in depth.

Thank you for reading and commenting on the manuscript.

I only have one question, regarding the calculation of the sample and the choice of the service, since methodologically the work is impeccable.

Page 2, lines 78~79

The minimum required sample size was estimated based on the suggestion that there be at least 200 participants for exploratory factor analysis [27].

Reviewer 2 Report

It was with keen interest that I read the validation study regarding a very significant, although still greatly underestimated task of a registered nurse which is health education. The study under review might contribute, providing scientific evidence, to the performance of this particular activity of nurses.   Comments regard the below mentioned issues: 

  1.     It would be beneficial if the authors of the study could address the criteria for including and excluding nurses from the study.
  2. .     It would be advisable to point out the selection criteria used for haemodialysis departments.

Author Response

Reviewer# 3: Comments to Author:

Author response to comments:

It was with keen interest that I read the validation study regarding a very significant, although still greatly underestimated task of a registered nurse which is health education. The study under review might contribute, providing scientific evidence, to the performance of this particular activity of nurses.   Comments regard the below mentioned issues: 

Thank you for reading and commenting on the manuscript.

It would be beneficial if the authors of the study could address the criteria for including and excluding nurses from the study.

Page 2, lines 79~82

The criteria for selecting the study participants were registered nurses working in the hemodialysis units of private clinics, general hospitals, and advanced general hospitals in six provinces of the Republic of Korea.

It would be advisable to point out the selection criteria used for haemodialysis departments.

Page 2, lines 82~83

The hemodialysis units were selected according to regional allocation and classifications of hospital types using a convenience sampling method.

Reviewer 3 Report

A Validation Study of the Korean Version of the Nurses’ Patient Education Questionnaire  

The manuscript was well written. I only have minor comment.

Participants:

Inclusion and exclusion criteria should be described. what about the sample size?

Measurements:

How was the initial draft developed? How was the process from 33 items to 26 items?

Was a pilot study conducted?

Author Response

Reviewer# 2: Comments to Author:

Author response to comments:

The manuscript was well written. I only have minor comment.

Thank you for reading and commenting on the manuscript.

Participants:

Inclusion and exclusion criteria should be described.

Page 2, lines 79~82

The criteria for selecting the study participants were registered nurses working in the hemodialysis units of private clinics, general hospitals, and advanced general hospitals in six provinces of the Republic of Korea.

what about the sample size?

Page 2, lines 86~87

The final sample size was 262, which satisfied the minimum number of participants for exploratory factor analysis.

Measurements:

How was the initial draft developed?

Page 3, lines 101~103

Nurse patient education competency was measured using the Nurses’ Patient Education Questionnaire. The initial items were developed based on the literature review and validated by the group of experts [25, 28].

How was the process from 33 items to 26 items?

The 33 item-Nurses’ Patient Education Questionnaire was refined to 26 item- Nurses’ Patient Education Questionnaire after item analysis ( 3 items were deleted with item–to–total scale correlations < .30 or > .70) and exploratory factor analysis (4 items with < 0.60 factor loading were deleted).

Page 5, lines 168~175

Three items were eliminated because of redundancy or lack of homogeneity with the construct: (1) I am uncertain of what patient information the physician should provide as opposed to myself as a nurse and what the content should be (0.29); (2) I am uncertain of what patient teaching the physician should provide as opposed to myself as a nurse and what the content should be (0.28); and (3) I know how to document patient teaching in the record (0.71). The remaining 30 items had corrected item–total correla-tion coefficients between 0.31and 0.68 which is acceptable range [30].

Page 6, lines 186~191

Four items with < 0.60 factor loading were deleted: (1) I know how to document patient information in the record (0.40); (2) I document learning objectives for patient teaching in the record (0.47); (3) After the patient has had a conversation with the physician, I check how the patient has understood the physician’s patient education (0.45); (4) I think patient teaching is an important nursing responsibility (0.45). Together, these 26 items accounted for 70.2% of the variance (Table 2).

Was a pilot study conducted?

Page3, lines 118~119

Three hemodialysis unit nurses participated in a pilot test of the Korean version Nurses’ Patient Education Questionnaire. No items were revised during this process.